# YesNo-Pro: A High-Performance Pointwise Reranking Algorithm Bridging Encoder-Decoder and Decoder-Only LLMs

## Abstract

Recent research has shown significant progress in the field of zero-shot text reranking for large language models (LLMs). Traditional pointwise approaches prompt the LLM to output relevance labels such as "yes/no" or fine-grained labels, but they have several drawbacks. Firstly, these prompts struggle to capture complex correlations between queries and passages and lack robustness for outputs not covered by predefined labels. Secondly, ranking scores rely solely on the likelihood of relevance labels, leading to potential noise and bias. Lastly, existing pointwise approaches are not supported by decoder-only LLMs, as ranking requires LLMs to output prediction probabilities. In response to these challenges, a novel pointwise approach called yesno-pro has been designed, which redefines both prompt design and score computation mechanisms to better align with the intrinsic nature of text reranking. Additionally, a comprehensive reranking framework based on LLM services has been proposed to support concurrent ranking calls and quickly adapt to any open-source decoder-only large models. Experimental results have demonstrated that this method outperforms existing pointwise and some pairwise/listwise methods on TREC19/20 and BEIR datasets, achieving the state-of-the-art performance. Due to its concurrency features, this work is applicable to practical applications with high real-time requirements.

## 1 Introduction

In recent years, LLMs(large language models) have significantly empowered an increasing number of domains. In this paper, we focus specifically on zero-shot text reranking. Compared to traditional methods, using LLMs to perform zero-shot text reranking demonstrates significant advantages due to their capability to understand knowledge across diverse domains. For example, utilizing LLMs for text reordering mitigates the necessity for repeated fine-tuning of small models in situations where knowledge is subject to frequent changes across different domains. This approach effectively reduces both resource expenditure and time. Recent works related to zero-shot LLM rankers can be categorized into three types: pairwise, listwise, and pointwise methods.

In pairwise approaches, LLMs are prompted with a query along with a pair of passages to perform ranking tasks(Qin et al., 2023)(Luo et al., 2024). Conversely, listwise approaches involve prompting the LLM with a query and a comprehensive list of passages(Sun et al., 2023). The two approaches share the following common points: (1) They require the design of sampling strategies, such as "sliding window" and "all pair", to obtain candidate passage sets that cover all passages of a query. These candidate passage sets can be structured as either pairs or lists; (2)The ranking of any given passage is inherently relative, being contingent upon comparisons with other passages. This leads to three drawbacks: (1) Due to the max token length limitation of LLMs, pairwise and listwise approaches cannot be scaled to long lists. (2) The required sampling strategies introduce additional time overhead. Particularly in pairwise methods, the total number of candidate passage sets often exceeds the number of original passages, leading to an increase in overall inference times. (3) The comparative nature prevents the two approaches from supporting concurrent invocations. These factors render pairwise and listwise approaches relatively slow, and limit their application in practical scenarios with high real-time requirements, thereby confining their use predominantly to research settings.

Existing pointwise rankers can be categorized into two classes: query generation and relevance generation. Query generation often ranks documents based on the query likelihood from LLM given the passage(Sachan et al., 2022a). With superior performance, relevance generation generally prompts the LLM to generate relevance labels given a query and a passage,such as "yes/no" or "0,1,2,3", and subsequently derives ranking score based on their likelihood. However, this approach suffer from these drawbacks: (1) Existing pointwise prompts are insufficient to capture the intrinsic relevance between queries and passages. This indicates that, although there may be some connection between the passage and the query, it does not provide actual assistance in answering or resolving the query. We will discuss this in detail in Section 2.1; (2) At times, the outputs of LLMs do not conform to predefined relevance labels, and current approaches fail to effectively mitigate this issue, lacking corresponding solutions; (3) Zero-shot text reranking is generally applied to reorder passages based on the ranking results from the first-stage, but existing methods do not fully leverage these first-stage ranking results, which constrains further improvements in ranking performance; (4) Current pointwise approaches are applicable to encoder-decoder LLMs but do not support decoder-only LLMs. Furthermore, there exists a lack of an efficient and deployable framework for text rankers using LLMs in the open-source domain.

A question naturally arises: Is it possible to propose a new zero-shot LLM reranking method that overcomes the shortcomings of pairwise/listwise and pointwise approaches while balancing both accuracy and efficiency?

In this work, we present a novel pointwise method: Yesno-Pro. This approach introduces a effective prompt template that guides LLMs in capturing the key correlations between queries and passages and employs a comprehensive scoring computation method that leverages the first-stage ranking results, mitigating biases and noise arising from reliance on a singular information source. Furthermore, we developed a VLLM-based text reranking framework that supports concurrent calls and significantly enhances the ranking speed. Our contributions can be summarized as follows:

1. We propose a new approach: Yesno-Pro. By optimizing the prompt and ranking score deriving methods, this approach substantially improves the performance and surpass other pointwise llm rankers and some pairwise/listwise approaches.

2. Yesno-Pro is the first pointwise approach capable of supporting both encoder-decoder and decoder-only LLMs.

3. We develop a LLM text reranking framework that can swiftly adapt to any LLM model supported by open VLLM, which support concurrent calls and demonstrates superior speed compared to all existing LLM text rankers.

## 2 MOTIVATION

The introduction has briefly outlined the drawbacks of existing listwise/pairwise and pointwise methods. This section will delve deeper into the fundamental reasons behind the suboptimal performance of current pointwise approaches, focusing on two key aspects: prompt design and ranking score computation, illustrated with concrete examples.

### 2.1 PROMPT DESIGN

Fig 1 lists the prompts in popular pointwise methods. Among these, prompts (a), (b), and (c) guide the large model to output relevance labels between queries and documents. However, such prompts may hinder LLMs to effectively capture the critical relevance between them. For instance, consider a user query such as "What are the market trends in the electric vehicles?" Document 1 states: "The market trend for traditional gasoline vehicles is gradually declining, influenced by electric vehicles and environmental policies, leading to shifts in consumer preferences." In contrast, Document 2 asserts: "Sales of electric vehicles have increased by 30% over the past year." In reality, Document 2 should be prioritized over Document 1, as it directly addresses the user's query. However, if relevance-based prompts are employed, the model might inaccurately prioritize Document 1 over Document 2, as it could focus on some irrelevant aspects of information.

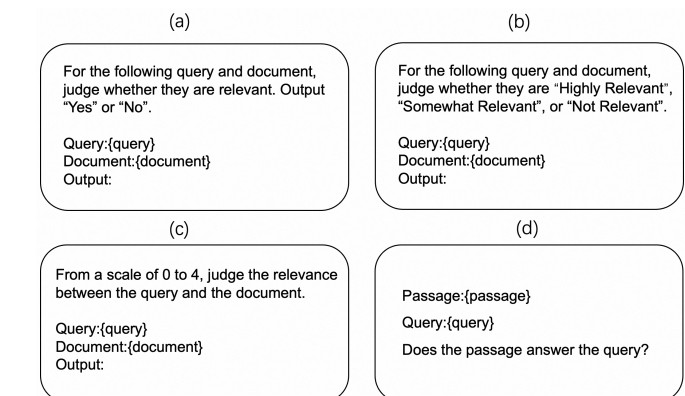

Figure 1: **Metric Learning Constraint Network**: First, X, the embeddings of all patches in a WSI, which are extracted through pretrained ResNet50, are linearly projected to obtain patch-level features $f_p$. Second, the patch-level features are aggregated by bilinear gated attention mechanism to obtain slide-level features $f_s$. And A is the learned attention score. Third, classification loss, center cluster loss and instance-level clustering loss are computed based on $f_s$, $f_p$ and A to train the whole network. Notice that

Certain prompts will ask LLMs whether the passage answer the query, as illustrated in prompt (d). However, there are instances in which the passage does not explicitly respond to the query but still contains some useful information.

In addition, LLMs sometimes fail to produce precise outputs. For example, if the query is "Is a Shiba Inu a pet?" and the passage is "In the dog sales market, a Shiba Inu costs 6,000 yuan.", under the current prompting templates, the LLM might respond with, "The two statements are not directly related, but dogs belongs to pets, so a Shiba Inu is a pet." rather than providing a binary answer of "yes/no" or other predefined relevance labels, which can potentially affect the calculation of ranking scores, ultimately influencing the performance of the final ranking.

## 2.2 RANKING SCORE COMPUTATION

Existing LLM rerankers are typically employed to refine the ranking of passages subsequent to the initial ranking stage. Popular pointwise approaches derive ranking scores based on the probabilities of relevant labels generated by LLMs. Inspired by (Luo et al., 2024), we identify a significant limitation of this approach: it does not effectively integrate information from the first-stage ranking. In order to avoid noise and bias, we should consider how to effectively integrate ranking results of the two stages to enhance overall ranking performance.

In addition, current pointwise approaches only support LLMs of encoder-decoder structur, but the majority of contemporary and advanced LLMs utilize a decoder-only structure. This limitation constrains the applicability of pointwise LLM rerankers across diverse contexts.

## 3 YESNO-PRO

Considering the aforementioned in-depth analysis of the suboptimal performance and usability challenges of existing pointwise approaches, we propose Yesno-Pro, which optimizes prompt design and ranking score computation.

### 3.1 PROMPT DESIGN

We designed the following prompt:

"Passage:{text} Query:{query} Does this passage contain the information needed to answer the question? Please respond directly with 'Yes' or 'No'."

In this prompt, "Does this passage contain the information needed to answer the question?" addresses the challenge that relevant-based prompts' inability to adequately guide LLMs to capture the critical relevance between passages and queries. It also mitigates the confusion that arises from "Does the passage answer the query?", as some passages may not directly respond to the query posed. Furthermore, the instruction "Please respond directly with 'Yes' or 'No'." effectively reduces the negative impact on the ranking results caused by the fact that LLMs do not always produce responses aligned with the predefined relevant labels.

We will conduct a detailed investigation into the effects of different prompts in Ablation Study.

## 3.2 RANKING SCORE COMPUTATION

Given a query q and its first-stage ranking results $D = \{(d_1, r_1), (d_2, r_2), ..., (d_n, r_n)\}$, where $d_i$ denotes the i-th passage, $r_i$ represents the score of $d_i$, and n is the total number of passages, the process of reranking is to prompt LLMs, derive ranking scores for each $(q, d_i)$ and subsequently perform sorting based on ranking scores.

For encoder-decoder LLMs, following traditional methods, we can caculate ranking scores by:

$$s_i = \frac{e^{p_{i,1}}}{e^{p_{i,1}} + e^{p_{i,0}}} \tag{1}$$

where,

$$p_{i,1} = LLM(Yes|q.d_i), p_{i,2} = LLM(No|q, d_i)$$

Given a query q and the i-th passage $d_i$, $p_{i,1}$ denotes the likeklihood that LLM outouts 'Yes' , and $p_{i,2}$ denotes the likelihood that LLM outputs 'No'. The ranking score $s_i$ is calculated using a softmax function.

For decoder-only LLMs, to overcome the limitation that pointwise approaches only support encoder-decoder LLMs, we optimized the vllm framework(Kwon et al., 2023), a widely used LLM server, to enable the LLMs to ouput not only generated tokens but also the logits corresponding to each token.

We assume the output of LLM is

$$S = \{(t_1, l_1), (t_2, l_2), ..., (t_N, l_N)\}$$

where N denotes the number of output tokens, $t_i$ denotes the i-th output token, and $l_i \in R^{1 \times K}$ denotes the logits corresponding to the i-th output token, here K is the vocabulary size.

Firstly, we need to identify the positions of 'Yes' and 'No' within the output tokens. There are two situations here. In the first scenario, the output contains either 'Yes' or 'No', and we denote the position of 'Yes' or 'No' within the output tokens as p. Then we use the logits $l_p \in R^{1 \times K}$ corresponding to token p for ranking score calculation. Let us denote the token id for 'Yes' as m and the token id for 'No' as n. Accordingly, the ranking score for the i-th passage is computed as follows:

$$s_i = \frac{e^{l_p[m]}}{e^{l_p[m]} + e^{l_p[n]}} \tag{2}$$

where $s_i$ is a value that lies within the interval [0, 1].

In the second scenario, If the output of the LLM contains neither "yes" nor "no," then we assign

$$s_i = 0.5 \tag{3}$$

In this way, for both encoder-decoder and decoder-only LLMs, we can derive ranking score $s_i$ for query q and passage $d_i$. However, scores derived from a single perspective are prone to bias and noise. Therefore, it is essential to incorporate the initial scores from the first stage to enhance the robustness. The final ranking score $S_i$ can be calculated using the following formula:

$$r_{max} = max\{r_1, r_2, ..., r_n\} \tag{4}$$

$$r_{min} = min\{r_1, r_2, ..., r_n\} \tag{5}$$

$$S_i = s_i(r_{max} - r_{min}) + r_{min} + \alpha * r_i \qquad (6)$$

In this context, the expression $s_i(r_{max} - r_{min}) + r_{min}$ serves to normalize the ranking score from the second stage to align with the interval of the initial ranking score $r_i$ from the first stage, thereby ensuring their additivity. The parameter $\alpha$ represents the weight factor that balances the ranking scores between the two stages. This factor can be adjusted in different datasets. In ablation study, we will investigate the impact of varying values of $\alpha$ on reranking performance.

## 4 EXPERIMENTS

### 4.1 COMPARISON OF RANKING PERFORMANCE

#### 4.1.1 DATASET AND METRICS

TREC and BEIR are two widely used benchmark datasets in text reranking tasks.

TREC encompasses a diverse array of topics and text types, including news articles, web content, email correspondence, and various forms of queries pertaining to these texts. For TREC, we use the test sets of passage reranking tasks in TREC-DL 19 and TREC-DL 20. Sharing the same MS MARCO v1 passage corpus, TREC-DL 19 contains 43 queries and TREC-DEC 2020 contains 55 queries.

BEIR (Benchmarking Information Retrieval)(Thakur et al., 2021) is a dataset designed for evaluating retrieval models across various domains and tasks. For BEIR, following previous work(Qin et al., 2023)(Zhuang et al., 2023)(Sun et al., 2023), we choose 7 test sets: Covid, Robust04, Touche, DBPedia, SciFact, Signal and News for evaluation.

For each query in both TREC and BEIR datasets, we re-rank the top 100 passages retrieved by BM25(Lin et al., 2021) and evaluate ranking performance using nDCG@ $\{1,5,10\}$ , which is under the same settings as previous works.

#### 4.1.2 BASELINES

We evaluate our Yesno-Pro on both encoder-decoder LLMs and decoder-only LLMs. For encoder-decoder LLMs, we use FLAN-T5-XL, FLAN-T5-XXL(Chowdhery et al., 2023) and FLAN-UL2(Tay et al., 2023). For decoder-only LLMs, we use Qwen2-7b and Qwen2-72b(Yang et al., 2024). We compare our method with below works, which are all zero-shot LLM rankers.

- RANKGPT: As a typical listwise approach, it is proposed in (Sun et al., 2023).
- UPR: As a query generation based pointwise approach, it is proposed in (Sachan et al., 2022b)
- RG: As a relevance generation based pointwise approach, it is proposed in (Liang et al., 2023)
- RG-S(0,4): As a fine-grained relevance generation based approach, it is proposed in (Zhuang et al., 2023)

#### 4.1.3 MAIN RESULTS

We conducted a comparative analysis of our Yesno-Pro and existing LLM rankers on the TREC and BEIR benchmarks, as illustrated in Table 1. Based on the results, we draw the following conclusions:

(1) For both encoder-decoder LLMs and decoder-only LLMs, when maintaining the same base model, Yesno-Pro significantly outperforms existing pointwise and listwise methods in terms of the NDCG@10 metric on both the TREC and BEIR datasets. The most notable improvement occurs with the qwen2-7b base model, where our method surpasses RankGPT by 5.23 points on BEIR and outperforms RG-S(0,4) by 4.37 points. Similarly, on the DL19 dataset, our approach exceeds RankGPT by 10.73 points and RG-S(0,4) by 3.64 points, while on the DL20 dataset, it outpaces RankGPT by 10.45 points and RG-S(0,4) by 7.12 points.

(2) On the TREC dataset, the top-performing model is Flan-UL2, whereas on the BEIR dataset, the leading model is Qwen2-72b.

Table 1: NDCG@10 on TREC and BEIR. All models re-rank the same BM25 top-100 passages. We mark the best performing models bold. In the table, 'q7b' refers to Qwen2-7b(Yang et al., 2024), 'q72b' refers to Qwen2-72b(Yang et al., 2024), 'f-xxl' refers to Flan-T5-XXL(Chowdhery et al., 2023), and 'f-ul2' refers to Flan-T5-UL2(Tay et al., 2023).

| Method | DL19 | DL20 | Covid | Robust | Touche | DBPedia | Scifact | Signal | News | BEIR(Avg) |
|---|---|---|---|---|---|---|---|---|---|---|
| **Baseline** | | | | | | | | | | |
| BM25 | 50.58 | 47.96 | 59.47 | 40.7 | **44.22** | 31.8 | 67.89 | 33.05 | 39.52 | 45.23 |
| **Existing LLM Rankers** | | | | | | | | | | |
| RankGPT(q7b) | 51.38 | 48.51 | 61.72 | 39.75 | 40.98 | 32.48 | 66.03 | 32.31 | 35.51 | 44.11 |
| RG-S(0,4)(q7b) | 58.47 | 51.84 | **75.44** | 46.29 | 23.35 | 35.58 | 65.83 | 24.77 | 43.55 | 44.97 |
| UPR(f-xxl) | 62 | 60.34 | 72.64 | 47.85 | 21.56 | 35.14 | **73.54** | 30.81 | 42.99 | 46.36 |
| RG(f-xxl) | 64.48 | 62.58 | 70.31 | 51.56 | 22.1 | 31.32 | 63.43 | 26.89 | 37.34 | 43.28 |
| UPR(f-ul2) | 58.95 | 60.02 | 70.69 | 47.52 | 23.68 | 34.64 | 71.09 | 30.33 | 41.78 | 45.68 |
| RG(f-ul2) | 64.61 | 65.39 | 70.22 | 53 | 24.67 | 30.56 | 64.74 | 29.68 | 43.78 | 45.24 |
| **Ours** | | | | | | | | | | |
| Yesno-Pro(q7b) | 62.11 | 58.96 | 72.65 | **58.22** | 35.66 | **36.88** | 70.46 | 31.28 | 40.21 | 49.34 |
| Yesno-Pro(q72b) | 62.49 | 62.62 | 71.64 | 57.2 | 43.5 | 36.74 | 71.61 | 32.59 | 43.06 | **50.91** |
| Yesno-Pro(f-xxl) | 65.97 | 63.96 | 70.64 | 54.03 | 40.18 | 32.9 | 64.83 | 32.74 | 39.51 | 47.83 |
| Yesno-Pro(f-ul2) | **66.92** | **66.18** | 71.65 | 54.95 | 39.23 | 32.91 | 69.94 | **33.46** | **46.77** | 49.84 |

(3) The performance of encoder-decoder LLMs and decoder-only LLMs is relatively comparable. However, decoder-only models benefit from compatibility with the open-source LLM inference framework VLLM, resulting in faster inference speed and better suitability for practical deployment in business contexts.

## 4.2 COMPARISON OF RANKING SPEED

### 4.2.1 EXPETIMENT SETUP

In this part, we select a representative method from each of the pointwise, pairwise, and listwise approaches, specifically focusing on PRP(Qin et al., 2023), RankGPT(Sun et al., 2023), and RG(Liang et al., 2023). We then compare their ranking speeds with our proposed method, YesNo-Pro. For PRP, we use the PRP-Sliding-10 ranking strategy. In the case of RankGPT, we employ the sliding window strategy with a window size of 20 and a step size of 10. TREC-DL19 and TREC-DL20 datasets are utilized for testing, and Qwen2-7b(Yang et al., 2024) serves as the base model. In our experiment, we first perform a validation for each dataset, recording the total elapsed time. We then compute the average processing time per query by dividing the total elapsed time by the total number of queries present in the dataset.

### 4.2.2 MAIN RESULTS

As illustrated in Table 2, YesNo-Pro demonstrates a substantial performance improvement, achieving speeds that are twice that of RG, 6.2 times faster than RankGPT, and an impressive 76 times faster than PRP.Our approach significantly faster than existing methods such as PRP, RankGpt, and RG, which represent popular pairwise, listwise, and pointwise techniques. This advantage can be attributed to two key factors. Firstly, Yesno-Pro belongs to the pointwise category, whereby the ranking of each passage is independent of others. This independence reduces the time consumption associated with additional sampling and comparison processes. Secondly, unlike other pointwise methods, such as RG, our method is tailored to leverage vllm, enabling accelerated inference and concurrent invocation of LLMs.

Table 2: Ranking Speed of different LLM Rankers.The unit of measurement is s/query. The best performing model is marked in bold.

| Method | DL19 | DL20 |
|--------|------|------|
| PRP | 131.3 | 132 |
| RankGPT | 10.7 | 10.6 |
| RG | 3.6 | 3.3 |
| Ours | **1.7** | **1.7** |

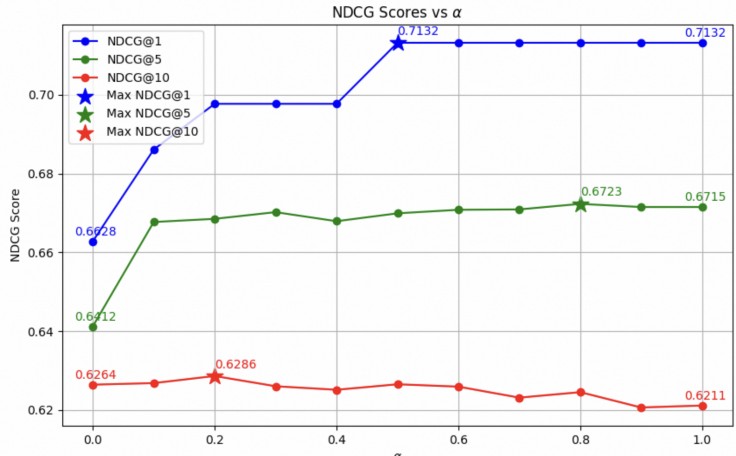

Figure 2: NDCG@$\{1, 5, 10\}$ on TREC-DL19 dataset with different $\alpha$. The blue line refers to NDCG@1, the green line refers to NDCG@5 and the red line refers to NDCG@10.

## 4.3 ABLATION STUDY

### 4.3.1 THE EFFECT OF WEIGHT FACTOR $\alpha$

To investigate the impact of incorporating the first-stage ranking scores in the final score computation, we conducted comparative experiments on the TREC-DL19 dataset. With Qwen2-7b serving as the base model, we plotted how the ranking performance(measured by NDCG@$\{1, 5, 10\}$), varies with changes in the weight balancing factor, denoted as $\alpha$. The values of $\alpha$ is within the range of $[0, 1]$, with a step size of 0.1 for each increment.

The results are shown in Figure 2. For NDCG@1, optimal ranking performance is achieved at $\alpha = 0.5$, resulting in a score of 0.7132. This represents a $5.04\%$ improvement compared to the scenario without the first-stage ranking score (where $\alpha = 0$). In the case of NDCG@10, the highest performance is observed at $\alpha = 0.8$, while the maximum score of 0.6286 for NDCG@10 occurs at $\alpha = 0.2$. These improvements can be attributed to the integration of scores from two stages, which effectively mitigates biases and noise that arise from reliance on single-source information.

### 4.3.2 THE EFFECT OF DIFFERENT PROMPTS

To investigate the impact of different prompts on ranking performance, we conducted comparative experiments with the three templates illustrated in Figure 3. Template 1 and template 2 are taken from existing pointwise approaches(Liang et al., 2023)(Zhuang et al., 2023). In this experiment, we use the qwen2-7b model as the base model and the TREC-DL19 dataset as our test set.

As shown in table3, the results indicate that template2 outperforms template1. This is because relevance generation based approaches often fail to capture the critical correlations between queries and passages, which is discussed in Section 2. In contrast, the approach of determining whether the passage contains the necessary information to address the query aligns more closely with the intrinsic requirements of the ranking task, thereby enhancing performance. Furthermore, the superi-

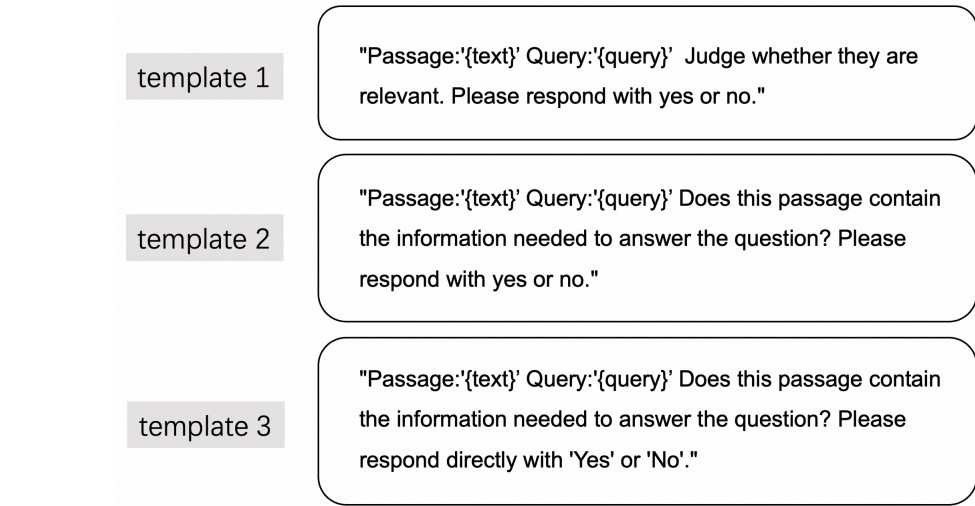

Figure 3: Three different prompt templates utilized in the experiment.

Table 3: Results of different prompts on the TREC-DL2019 dataset.The best performing template is marked in bold.

| prompts | DL19 | | |
|---|---|---|---|
| | NDCG@1 | NDCG@5 | NDCG@10 |
| template1 | 68.99 | 64.56 | 61.12 |
| template2 | 67.44 | 66.01 | 61.54 |
| template3 | **71.32** | **67.13** | **62.09** |

ority of template3 over template2 suggests that the instruction "'Please respond directly with 'Yes' or 'No'" effectively reduces the occurrence of tokens outside predefined labels generated by the LLMs, thereby enhancing the ranking performance.

## 5 CONCLUSION

In this paper, we introduce YesNo-Pro, a novel pointwise ranking method. By optimizing both the prompt design and the ranking score derivation, our approach significantly surpasses existing pointwise and listwise methods, achieving state-of-the-art results. Furthermore, we have enhanced the VLLM framework to support YesNo-Pro, substantially accelerating the ranking speed and making it the fastest among current pointwise, pairwise, and listwise methods. This adaptation offers additional advantages, such as bridging encoder-decoder and decoder-only LLMs, and enabling the algorithm to support concurrent calls. Consequently, it can be rapidly adapted to any open-source LLMs and efficiently deployed in practical scenarios. We are looking forward to further study in highly effective and rapid pointwise LLM rerankers.

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
