# OpenReview forum: "YESNO-PRO: A HIGH-PERFORMANCE POINTWISE RERANKING ALGORITHM BRIDGING ENCODERDECODER AND DECODER-ONLY LLMS"
_ICLR.cc/2025/Conference — ICLR 2025 Conference Withdrawn Submission_

### Official Review · Reviewer_SAcG · 2024-11-02

**Soundness:** 3
**Presentation:** 3
**Contribution:** 2
**Rating:** 3
**Confidence:** 4

**Summary:**

The paper proposes an improvement to how LLM rankers can be used for document reranking. In this direction they propose:
- A new prompt designed to make the LLM focus on whether the content in the passage can be used to answer the query, instead of focusing on whether the passage/document answers the query.
- The paper also proposes a way to combine the scores from the retrieval stage and claim that this is better than using the score for the ranking stage alone. It scales the scores so that they are in comparable range before combining them.
- Ablations on different prompt strategies are provided to support effectiveness of the new prompt design.
- Ablations on weight of scores from the two retrieval/ranking stages.
- Ablations on different models, and ranking strategies (pointwise, listwise).

**Strengths:**

- A valuable proposal to change the strategy to prompt design for retrieval/ranking since often we don't want just passages/documents that answer a given query, but provide information that can be used to answer the query.
- Proposal to rescale the scores before combining the scores from the two stages.
- Good set of ablations as described in the summary section.

**Weaknesses:**

- The abstract mentions "Traditional pointwise approaches
prompt the LLM to output relevance labels such as yes/no or fine-grained labels, but they have several drawbacks.", but then the paper proceeds to use the same strategy of using the logits corresponding to the yes/no labels which is the common way. There is no proposal to improve upon this.
- It is unclear from the experiments whether combining the scores from the two stages is really beneficial. From the ablation in table 2, we see that the first stage ranker is better for NDCG@10, while the second stage ranker is better for NDCG@{1,5}. The paper mentions that \alpha can be tuned across datasets, but the ablation is for different NDCG metrics on the same dataset.
- Paper mentions "Current pointwise approaches are applicable to encoder-decoder LLMs but do not support decoder- only LLMs.". However, later on, the paper again mentions "For decoder-only LLMs, to overcome the limitation that pointwise approaches only support encoder- decoder LLMs, we optimized the vllm framework(Kwon et al., 2023), a widely used LLM server, to enable the LLMs to ouput not only generated tokens but also the logits corresponding to each token.". So this is not a limitation of the decoder only model, but a limitation of the serving stack used by the authors. Infact, after stating that "For encoder-decoder LLMs, following traditional methods, we can caculate ranking scores", they use the same traditional methods for decoder only LLMs as well with no novelty in the modeling.
- "For decoder-only LLMs, to overcome the limitation that pointwise approaches only support encoder- decoder LLMs, we optimized the vllm framework" --> This should be updated to "we modified the vllm framework" instead since there is no optimization involved and the only change here is to return the logits as part of the API response.
- line 206: "If the output of the LLM contains neither ”yes” nor ”no,”" --> how can the output of the LLM sometimes contain yes/no tokens and sometimes miss them. Are you returning only top k token logits from the server? If yes, Won't it be better to just return the logits corresponding to the yes and no tokens?
- "S_{i} = s_i (r_{max} − r_{min}) + r_{min} + α ∗ r_{i}" --> Instead of rescaling the output of stage 2 in the same range as the output of stage1, it might be better to scale the output of stage1 in the range [0,1] as well making the formula much cleaner.
- In ablations, why are pointwise and listwise baselines included, but no pairwise even after mentioning about it in the paper?
- For the model comparisons, it would be better to mention the size of each model used in terms of number of parameters.
- line 409: "effectively reduces the occurrence of tokens outside predefined labels" --> It might be better to quantify this so that we can segregate the gains from the two proposed changes in the prompt, which are 1) shift from passage answering the query, and 2) including "directly" in the prompt.
- Even though having "directly" in the prompt helps in the extraction of answer from the model's response, CoT (Chain of Thoughts -- https://arxiv.org/pdf/2201.11903) emphasizes that letting the model reason about its response before generating the answer is much more performant and is widely used knowledge. A commonly used strategy is to ask the model to reason about the answer it is about to generate, but then end its response in a very specific format "<reason>.. Hence, the final answer is <yes/no>".
- line 163: relevant-based --> relevance-base
- "addresses the challenge that relevant-based promps' inability" --> This part can be reworded for clarity.

**Questions:**

See weakness section.

---

### Official Review · Reviewer_aud4 · 2024-11-03

**Soundness:** 2
**Presentation:** 2
**Contribution:** 2
**Rating:** 3
**Confidence:** 5

**Summary:**

This paper introduces an LLM prompting method for pointwise text reranking. It compares different prompt templates and designs a fusion function to combine reranker scores with first-stage retriever scores. The method is applied to both encoder-decoder and decoder-only LMs. Evaluation on TREC-DL and BEIR shows improvement over other pointwise prompting approaches.

**Strengths:**

- This paper proposes a new pointwise prompting technique that outperforms previous methods.

- The proposed method has been implemented for both encoder-decoder and decoder-only language models (LMs) and has been accelerated using vLLM.

**Weaknesses:**

- The proposed method involves only minor modifications to the prompt wording, which lacks novelty. And there is existing work on automatic prompt optimization for LLM rerankers (https://arxiv.org/pdf/2406.14449).

- This paper claims that “pointwise approaches only support encoder-decoder LLMs”, which is incorrect, as RankGPT (https://arxiv.org/abs/2304.09542) already applies pointwise prompting using the GPT API. And the use of vLLM is a common practice for LLM deployment and can also be applied to other baselines.

- This paper lacks comparison to state-of-the-art prompting methods such as setwise ranking (https://arxiv.org/abs/2310.09497), graph-based ranking (PRP-Graph), or tournament-based ranking (https://arxiv.org/abs/2406.11678).

- There are some formatting issues: the caption of Figure 1 appears to be an uncleaned template, and Table 1 exceeds the right margin of the page.

**Questions:**

None

---

### Official Review · Reviewer_j191 · 2024-11-03

**Soundness:** 1
**Presentation:** 2
**Contribution:** 1
**Rating:** 1
**Confidence:** 4

**Summary:**

This paper proposed a new LLM-for-ranking model called YesNo-Pro. The authors write a new prompt and design a score computation that includes scores from first-stage retrieval process. However, the designed prompt is not much different from previous prompts and does not provide new insights. The idea of using the scores from first-stage retrieval is also quite common. Some statements in the paper are confusing or even wrong.

**Strengths:**

Many experiments are conducted.

**Weaknesses:**

1. The designed pointwise-ranking prompt does not significantly differ from existing prompts and offers no additional insights. There are also papers([1,2]) discussing the prompt designs for zero-shot LLM-based rankers. The experiments in Section 4.3.2 also infer that the influences of different prompts on ranking performances are low.

2. The idea of using scores from retrieval stages is also not novel. Many previous works([4,5]) in hybrid retrieval have discussed it.

3. Some statements in this paper are confusing or even wrong.
    - The authors state that ``Pointwise approaches are not supported by decoder-only LLMs``. This is wrong, many approaches([1,3]) apply decoder-only LLMs for pointwise ranking.
    - The authors state that ``At times, the outputs of LLMs do not conform to predefined relevance labels, and current approaches fail to effectively mitigate this
issue, lacking corresponding solutions``. This is wrong, we can still calculate the probabilities by resorting to the logits.
    - The example in Section 2.1 in line 101 is very confusing. Why LLM based on relevance-based prompts cannot yield correct results? Authors need to explain why LLMs cannot understand ``relevance`` in detail instead of simply stating it.

4. The authors didn't show how much improvements of YesNo-Pro model come from the ranking scores in the first-stage retrieval models. To my knowledge, most previous LLM-for-ranking models don't use scores in first-stage retrieval, which makes the comparison unfair.

5. The typos in the paper strongly affect the reading. For example, the description of Figure 1 is totally irrelevant to the contents in it.

[1] Sun, S., Zhuang, S., Wang, S., & Zuccon, G. (2024). An Investigation of Prompt Variations for Zero-shot LLM-based Rankers. SIGIR 2024.
[2] Zhuang, H., Qin, Z., Hui, K., Wu, J., Yan, L., Wang, X., & Bendersky, M. (2023). Beyond yes and no: Improving zero-shot llm rankers via scoring fine-grained relevance labels.
[3] Ma, X., Wang, L., Yang, N., Wei, F., & Lin, J. (2024, July). Fine-tuning llama for multi-stage text retrieval.
[4] Bruch, S., Gai, S., & Ingber, A. (2023). An analysis of fusion functions for hybrid retrieval. ACM Transactions on Information Systems, 42(1), 1-35.
[5] Kuzi, S., Zhang, M., Li, C., Bendersky, M., & Najork, M. (2020). Leveraging semantic and lexical matching to improve the recall of document retrieval systems: A hybrid approach.

**Questions:**

See the weaknesses.

---

### Official Review · Reviewer_XzZ8 · 2024-11-04

**Soundness:** 2
**Presentation:** 1
**Contribution:** 1
**Rating:** 3
**Confidence:** 4

**Summary:**

This paper presents a new approach called "yesno-pro" for zero-shot text reranking. The authors claim this new method can improve prompt design and support both encoder-decoder and decoder-only models. Experiments on TREC19/20 and BEIR datasets show this method can achieve better ranking results compared to other baseline methods.

**Strengths:**

1. The paper studies an important and practical problem
2. The proposed method, although standard, is reasonable

**Weaknesses:**

The paper has very limited contributions. The prompt design is standard and the way to prompt LLMs is also the same as the previous approaches. The re-ranking idea (in equations 4-6) is just a score merging, which also requires an existing pre-ranking stage.

The paper presentation is terrible, lots of typos, grammatical and formatting issues, Just list a few below:

1. Line 044 ”sliding window” and ”all pair”, the quotations are wrong
2. Line 057 “... passage,such as ”yes/no””, no space between comma
3. Line 059, “However, this approach suffer from these drawbacks”, "suffer " -> “suffers”
4. Figure 1 caption is completely wrong
5. Table 1 is terribly formatted

Overall, I think the paper is clearly below the acceptance threshold.

**Questions:**

1. Can your method be applied to first-stage ranking?
2. How is this method improve (other than the score merging) improves the previous LLM-based text ranking methods in your related work section?

---

### Note · Authors · 2024-11-18

I have read and agree with the venue's withdrawal policy on behalf of myself and my co-authors.